# NR4A1 Ligands as Potent Inhibitors of Breast Cancer Cell and Tumor Growth

**DOI:** 10.3390/cancers13112682

**Published:** 2021-05-29

**Authors:** Keshav Karki, Kumaravel Mohankumar, Abigail Schoeller, Gregory Martin, Rupesh Shrestha, Stephen Safe

**Affiliations:** 1Department of Veterinary Physiology and Pharmacology, Texas A&M University, College Station, TX 77843, USA; kkarki@cvm.tamu.edu (K.K.); kmohankumar@cvm.tamu.edu (K.M.); aschoeller@cvm.tamu.edu (A.S.); gmartin@cvm.tamu.edu (G.M.); 2Department of Biochemistry and Biophysics, Texas A&M University, College Station, TX 77843, USA; rshrestha@tamu.edu

**Keywords:** NR4A1, breast cancer, ligands, inhibition

## Abstract

**Simple Summary:**

Bis-indole derived (CDIMs) bind the orphan nuclear receptor 4A1 (NR4A1) and inhibit NR4A1-regulated cancer cell and tumor growth. In this study a subset of 3,5-disubstituted phenyl CDIM compounds that bound NR4A1 were investigated in a breast cancer model. All of these analogs were potent inhibitors of breast tumor growth in a xenograft model using MDA-MB-231 cells at doses ≤ 1 mg/kg/d.

**Abstract:**

Nuclear receptor 4A1 (NR4A1, Nur77, TR3) is more highly expressed in breast and solid tumors compared to non-tumor tissues and is a pro-oncogenic factor in solid tumor-derived cancers. NR4A1 regulates cancer cell growth, survival, migration, and invasion, and bis-indole-derived compounds (CDIMs) that bind NR4A1 act as antagonists and inhibit tumor growth. Preliminary structure-binding studies identified 1,1-bis(3′-indolyl)-1-(3,5-disubstitutedphenyl)methane analogs as NR4A1 ligands with low K_D_ values; we further investigated the anticancer activity of the four most active analogs (K_D_’s ≤ 3.1 µM) in breast cancer cells and in athymic mouse xenograft models. The treatment of MDA-MB-231 and SKBR3 breast cancer cells with the 3-bromo-5-methoxy, 3-chloro-5-trifluoromethoxy, 3-chloro-5-trifluoromethyl, and 3-bromo-5-trifluoromethoxy phenyl-substituted analogs decreased cell growth and the expression of epidermal of growth factor receptor (EGFR), hepatocyte growth factor receptor (cMET), and PD-L1 as well as inhibited mTOR phosphorylation. In addition, all four compounds inhibited tumor growth in athymic nude mice bearing MDA-MB-231 cells (orthotopic) at a dose of 1 mg/kg/d, which was not accompanied by changes in body weight. These 3,5-disubstituted analogs were the most potent CDIM/NR4A1 ligands reported and are being further developed for clinical applications.

## 1. Introduction

It is estimated that in 2020, there will be 279,100 new cases of breast cancer in the United States and 42,690 deaths from this disease [1]. Among women, breast cancer is the second leading cause of cancer deaths; however, mortality from breast cancer has decreased due to early detection and improved therapies. Therapeutic regimens for the treatment of breast cancer are highly variable and depend on the tumor type and stage [2]. Most early-stage breast cancers are estrogen receptor α—positive and epidermal growth factor receptor 2 (EGFR2, HER2) negative [3], and are treated with antiestrogens such as tamoxifen and/or aromatase inhibitors (e.g., anastrozole), which block the ERα-mediated function and inhibit 17β-estradiol synthesis, respectively [4]. Breast cancer patients with tumors that overexpress the epidermal growth factor receptor 2 (EGFR2, HER2) are successfully treated with the antibody Herceptin that targets and inactivates HER2 [5]. Patients with triple negative breast tumors that do not express HER2, ERα, or the progesterone receptor are more aggressive and highly invasive, and are treated with drug cocktails that include cytotoxic drugs, which also cause serious toxic side effects [6]. The orphan nuclear receptor 4A1 (NR4A1, TR3, Nur77) is overexpressed in tumors from patients with breast and several other cancers and is a negative prognostic factor for patient survival [7,8,9,10,11,12,13,14,15,16,17]. A survey of nuclear receptor expression in mammary tumors reported that NR4A1 was overexpressed in both ER-positive and ER-negative tumors but was inversely expressed with a higher histologic grade [14]. Other studies on different groups of breast cancer patients gave variable prognostic values for NR4A1 [14,15,17]. Functional studies on solid tumor-derived cancer cells have demonstrated that NR4A1 regulates genes and pathways associated with cell proliferation, survival, migration/invasion, and TGFβ-induced invasion (rev. in [18]). Thus, NR4A1 ligands that act as antagonists represent a novel class of anticancer agents for patients expressing this receptor. Studies in this laboratory initially identified 1,1-bis(3′-indolyl)-1-(p-hydroxyphenyl)methane (CDIM8) and the 4-carboxymethyl phenyl analog as NR4A1 ligands that act as antagonists and inhibit NR4A1-dependent pro-oncogenic pathways and genes in breast and other cancer cell lines [18,19,20,21,22,23,24]. CDIM8 was used as a prototypical NR4A1 ligand and as a scaffold for generating more potent ligands. CDIM8 binds NR4A1, and in breast cancer cells CDIM8 inhibits NR4A1-dependent transactivation and mimics the effects of an NR4A1 knockdown [20,21,22,23,24]. CDIM8 also antagonizes one or more NR4A1-regulated pro-oncogenic pathways or genes in several different breast cancer cell lines including HS578T, SUM159, MDA-MB-231, SKBR3, 4T1 (mouse), and MCF-7 cells [20,21,22,23,24].

The potency of CDIM8 and the 4-carboxymethylphenyl analog as inhibitors of tumor growth in vivo is in the 20–40 mg/kg/d range [13,20,25,26,27], and pharmacokinetic studies for CDIM8 show that this compound can rapidly be metabolized (conjugated) [28]. In contrast, a series of buttressed analogs of CDIM8 containing 3′- and 5′-substituents inhibited the growth, survival, and migration/invasion of ER-negative MDA-MB-231 and SKBR3 cells, and also mammary tumor growth in athymic nude mice bearing MDA-MB-231 cells (orthotopically) in the 2–5 mg/kg/d range of doses [21,22]. Moreover, one of these “buttressed analogs” of CDIM8 (3′-chloro-5-methoxy) also inhibited mammary tumor growth and enhanced immune surveillance in a syngeneic mouse model at a dose of 2.5 mg/kg/d [22]. Ongoing structure-receptor binding studies among a series of bis-indole-derived analogs confirmed the relatively high binding affinities of 3,5-disubstituted analogs of CDIM8 (containing the 4′-hydroxylphenyl group). In this screening study, we also investigated the assumed requirement of a 4′-hydroxy group for the binding of CDIMs to NR4A1. We observed in a series of 3′,5′-disubstitutedphenyl CDIM analogs that did not contain a 4′hydroxy group that these compounds also bound NR4A1. In this study, we investigated the in vitro and in vivo activity of a series of 3′,5′-disubstitutedphenyl CDIM analogs, which exhibited the highest binding affinity (K_D_) for NR4A1. These included the 3-chloro-5-trifluoromethyl (3-Cl- 5-CF_3_), 3-bromo-5-methoxy (3-Br-5-OCH_3_), 3-chloro-5-trifluoromethoxy (3-Cl-5-OCF_3_), and 3-bromo-5-trifluoromethoxy (3-Br-5-OCF_3_) substituted phenyl ring analogs. The results demonstrate that all four compounds inhibited SKBR3 and MDA-MB-231 cell growth, while in vivo studies showed that their IC_50_ values for tumor growth inhibition were <1 mg/kg/day.

## 2. Materials and Methods

### 2.1. Cell Lines, Antibodies, and Reagents

The CDIM 3,5-disubstituted phenyl analogs 1,1-bis(3′-indoly)-1-(3-bromo-5-methoxyphenyl)methane (3-Br-5-OCH_3_), 1,1-bis(3′-indolyl)-1-(3-chloro-5-trifluoromethoxyphenyl)methane (3-Cl-5-OCF_3_), 1,1-bis(3′-indolyl)-1-(3-chloro-5-trifluoromethylphenyl)methane (3-Cl-5-CF_3_), and 1,1-bis(3′-indolyl)-1-(3-bromo-5-trifluoromethoxyphenyl)methane (3-Br-5-OCF_3_) were synthesized by condensation of indole (2 mol equivalents) and the corresponding aldehyde (1 mol equivalent), as described [29]. The aldehydes included 3-bromo-5-methoxybenzaldehyde (Sigma-Aldrich, St. Louis, MO, USA), 3-chloro-5-trifluoromethoxybenzaldehyde (Alfa Aesar, Ward Hill, MA, USA), 3-chloro-5-trifluoromethylbenzalldehyde (Alfa Aesar), 3-bromo-5-trifluoromethoxybenzaldehyde (Alfa Aesar), and indole was obtained from Sigma-Aldrich. The compounds were >98% pure, as determined by LC-MS [29]. Human mammary breast cancer MDA-MB-231 and SKBR3 cell lines were purchased from American Type Culture Collection (Manassas, VA, USA). MDA-MB231 and SKBR3 were grown in Dulbecco’s Modified Eagle’s Medium (DMEM), supplemented with 10% FBS. Both cell lines were maintained at 37 °C in the presence of 5% CO_2_, and the solvent (dimethyl sulfoxide, DMSO) used in the experiments was ≤0.1%. EGFR and mTOR antibodies were obtained from Santacruz Biotechnology (Santacruz, CA, USA), c-Met, p-mTOR, anti-rabbit Alexa Fluor conjugated 488, anti-mouse Alexa Flour conjugated, and PD-L1 antibodies were obtained from Cell Signaling (Boston, MA, USA), NR4A1 antibody was obtained from Abcam (Cambridge, MA, USA), and β-actin antibody was obtained from Sigma Aldrich. Chemiluminescence reagents (Immobilon Western) for western blot imaging were purchased from Millipore (Billerica, MA, USA). XTT cell viability kit was obtained from Cell Signaling (Boston, MA, USA). Luciferase reagent was purchased from Promega (Madison, WI, USA). Lipofectamine 2000 reagent was obtained from Invitrogen (Carlsbad, CA, USA), and Matrigel was obtained from Corning (Tewksbury, MA, USA).

#### 2.1.1. Nuclear Magnetic Resonance Spectra and LC–MS Analysis

^1^H NMR spectra were measured on a Bruker Avance spectrometer (400 MHz) and are reported in ppm (s = singlet, d = doublet, t = triplet, q = quartet, m = multiplet, br = broad; integration; coupling constant(s) in Hz), using TMS as an internal standard (TMS at 0.00 p-pm) in CDC1_3_. Liquid chromatogram mass spectra (LCMS) were obtained on SHIMADZU2010 EV using methanol as solvent. All compounds were >98% pure by LC–MS. NMR spectra included the following results. 3-Cl-5-CF_3_: ^1^H NMR (400 MHz, CDCl_3_) δ 7.95 (s, 2H), 7.53 (s, 1H), 7.49 (s, 1H), 7.47 (s, 1H), 7.39–7.32 (m, 4H), 7.19 (ddd, *J* = 8.1, 6.8, 0.9 Hz, 2H), 7.03 (ddd, *J* = 7.9, 6.9, 0.8 Hz, 2H), 6.65 (dd, *J* = 2.6, 1.0 Hz, 2H), 5.92 (s, 1H). 3-Cl-5-OCF_3_: ^1^H NMR (400 MHz, CDCl_3_) δ 7.95 (s, 2H), 7.44 (m, 1H), 7.42–7.31 (m, 3H), 7.22–7.16 (m, 2H), 7.14 (s, 1H), 7.08–6.99 (m, 2H), 6.67 (dd, *J* = 2.5, 0.8 Hz, 2H), 5.87 (s, 1H). 3-Br-5-OCH_3_: ^1^H NMR (400 MHz, CDCl_3_) δ 7.91 (s, 2H), 7.40–7.34 (m, 4H), 7.22–7.13 (m, 2H), 7.10 (t, *J* = 1.7 Hz, 1H), 7.06–6.97 (m, 2H), 6.90 (t, *J* = 2.0 Hz, 1H), 6.84 (m, 1H), 6.68 (dd, *J* = 2.6, 1.08 Hz, 2H), 5.81 (s, 1H), 3.71 (s, 3H). 3-Br-5-OCF_3_: ^1^H NMR (400 MHz, CDCl_3_) δ 7.90 (s, 2H), 7.40–7.31 (m, 4H), 7.29–7.26 (m, 1H), 7.20–7.17 (m, 2H), 7.10 (s, 2H), 7.05–6.98 (m, 2H), 6.64 (d, *J* = 1.8 Hz, 2H), 5.86 (s, 1H).

#### 2.1.2. Computation-Based Molecular Modeling

Molecular modeling studies were conducted using Maestro (Schrödinger Release 2020-1, Schrödinger, LLC, New York, NY, USA, 2020). The version of Maestro used for these studies was licensed to the Laboratory for Molecular Simulation (LMS), a Texas A&M University core-user facility for molecular modeling and is associated with the Texas A&M University High Performance Research Computing (HPRC) facility (College Station, TX 77843, USA). All Maestro-associated applications were accessed via the graphical user interface (GUI) VNC interactive application through the HPRC Ada OnDemand portal. The crystal structure coordinates for the human orphan nuclear receptor NR4A1 ligand-binding domain (LBD) [30] were downloaded from the Protein Data Bank (https://www.rcsb.org, accessed on 5 September 2020; PDB ID 3V3Q). The human NR4A1 LBD crystal structure was prepared for ligand docking utilizing the Maestro Protein Preparation Wizard; restrained minimization of the protein structure was performed utilizing the OPLS3e force field. Each ligand’s (3-Br-5-OCH_3_, 3-Cl-5-OCF_3_, 3-Cl-5-CF_3_, or 3-Br-5-OCF_3_) three-dimensional structure was prepared for docking utilizing the Maestro LigPrep, again using the OPLS3e force field. Maestro Glide [31,32,33] was utilized with the default settings to dock each prepared ligand to the prepared protein, predict the lowest energy ligand-binding orientation, and calculate the predicted binding energy in units of kcal/mol.

### 2.2. Cell Viability Assay

Cells were plated in a 96-well plate at a density of 10,000 per well with DMEM containing 10% charcoal-stripped FBS. Cells were treated with DMSO (solvent control) and different concentrations of 3Br-5OCH_3_, 3Cl-5OCF_3_, 3Cl-5CF_3_, and 3-Br-5OCF_3_ with DMEM containing 2.5% charcoal-stripped FBS for 24 h. After treatment, 25 µL (XTT with 1% of electron coupling solution) was added to each well and incubated for 4 h, as outlined in the manufacturer’s instruction (Cell Signaling, Boston, MA, USA). Absorbance was measured at a wavelength of 450 nm in a 96-well plate reader after incubation for 4 h in 5% CO_2_ at 37 °C. The effects of the CDIMs on cell growth in MDA-MB-231 and SKBR3 cells after the knockdown of NR4A1 by RNA interference was determined essentially, as described [23].

### 2.3. Western Blot Analysis

Whole-cell lysates from MDA-MB-231 and SKBR3 were analyzed by western blots, as described [21,22]. Equal amounts of protein were separated in 8% and 10% SDS-PAGE and transferred to a polyvinylidene difluoride (PVDF) membrane. PVDF membranes were incubated overnight at 4 °C, with primary antibodies in 5% skimmed milk or 5% BSA and incubated for 2–3 h with secondary antibodies conjugated with HRP. Membranes were then exposed to an HRP-substrate and immune-reacted proteins were detected with a chemiluminescence reagent using a Kodak image developer. β-actin was used as a reference loading control.

### 2.4. Plasmid

The Flag-tagged full-length FLAG-NR4A1 expression plasmids were constructed by inserting PCR-amplified full-length NR4A1 (amino acids 1–599) into the *Eco*RI/*Bam*HI site and C-terminal NR4A1 (amino acids 67–599) and N-terminal NR4A1 (amino acids 1–354) into the *Eco*RI/*Kpm*I site of the p3XFLAG-CMV-10 expression vector (Sigma-Aldrich). NBRE_x3_-luc was generously provided by Dr Jacques Drouin (University of Montreal, Montreal, QC, Canada). All other plasmids used in this study were previously described [21,22].

### 2.5. Luciferase Reporter Assay

MDA-MB-231 and SKBR3 (10^5^) per well were plated on 12-well plates in DMEM with 2.5% charcoal-stripped FBS and 0.22% sodium bicarbonate, respectively. After 24 h growth, various amounts of DNA [i.e., GAL4-NR4A1 (500 ng), UAS-Luc (50 ng), and β-gal (50 ng) or Flag-NR4A1, NBRE-Luc, and β-gal (50 ng)] were transfected into each well by Lipofectamine 2000 reagent (Invitrogen, Carlsbad, CA, USA), according to the manufacturer’s protocol. After 5–6 h of transfection, cells were treated with solvent (DMSO) or the indicated concentration of compounds for 24 h. Cells were then lysed using a freeze–thaw protocol and 30 μL of cell extract was used for luciferase and β-gal assays. LumiCount (Packard, Meriden, CT, USA) was used to quantify luciferase and β-gal activities. Luciferase activity values were normalized against corresponding β-gal activity values as well as protein concentrations determined by Bradford method.

### 2.6. NR4A1 Binding Assay

Recombinant LBD of NR4A1 was incubated with different concentrations of ligand to obtain bind curves and K_D_ values. Binding was determined by tryptophan fluorescence spectra at 285 nm (excitation slit width = 5 nm) and an emission wavelength range of 300–420 nm (emission slit width = 5 nm). Ligand binding affinity (K_D_) to NR4A1 was determined by measuring NR4A1 tryptophan fluorescence intensity at an emission wavelength of 330 nm [19]. The 4 compounds used in this study did not interfere with the assay, whereas this has previously been observed with some CDIMs [19].

### 2.7. Xenograft Study

Female athymic nu/nu mice (4–6 weeks old) were purchased from Harlan Laboratories (Houston, TX, USA). MDA-MB-231 cells (5 × 10^6^) were harvested in 100 µL of DMEM and suspended in ice-cold Matrigel (1:1 ratio) and orthotopically transplanted into the mammary fat pads region of mice, as described [21,22]. After one week, mice were divided into five groups of 5 animals each. The first group received 100 µL of vehicle (corn oil), and the other groups received an injection of 1 mg/kg/day of the indicated compounds in 100 µL volume of corn oil by i.p. for 3 weeks. All mice were weighed once a week over the course of the treatment to monitor changes in body weight. After 3 weeks of treatment, mice were sacrificed, tumor weights were determined, and tumor lysates for control and treated groups were obtained for western blot analysis. All animal studies were carried out according to the procedures approved by the Texas A&M University Institutional Animal Care and Use Committee and the specific AUP number is 2020-138, which runs to 23 June 2023.

### 2.8. Statistical Analysis

An unpaired Student *t*-test or one-way ANOVA was used for statistical analysis. The unpaired Student *t*-test was used to calculate statistical significance for animal experiment results and luciferase assay to compare means between control and treatment, whereas the one-way ANOVA was used to measure significance between control and multiple treatment groups for cell viability results. In order to confirm the reproducibility of the data, the in vitro cell culture experiments were performed at least three independent times, and results were expressed as means ± SD. *p*-values less than 0.05, were statistically significant. For in vivo studies, 5 animals were used for each treatment group.

## 3. Results

### 3.1. CDIM—NR4A1 Binding and Transactivation

The interactions of bis-indole-derived CDIM compounds containing a 3,5-disubstituted phenyl group with the ligand-binding domain of NR4A1 were investigated by determining the fluorescence quenching of Trp in the ligand-binding pocket. The binding curves for the 3-Br-5-OCH_3_, 3-Cl-5-OCF_3_, 3-Cl-5-CF_3_, and 3-Br-5-OCF_3_ analogs are illustrated in Figure 1A–D and the K_D_ values of 1.8, 2.3, 3.1, and 2.0 µM, respectively, were similar for all 4 compounds. Docking each ligand to the NR4A1 LBD utilizing the Schrodinger Maestro, as described above, resulted in similar binding orientations for each ligand within the LBD binding pocket. For each ligand, the 3,5-disubstituted phenyl ring was oriented toward the solvent-exposed outer portion of the binding pocket. Calculation of the predicted binding energy also yielded similar values of −4 to −5 kcal/mol for each ligand. However, energy minimization yielded some differences in the predicted protein–ligand intermolecular interactions. Arg 184 in the NR4A1 LBD (Arg 515 in full-length NR4A1) was predicted to form a pi-cation interaction with one of the five-membered pyrrole rings in 3-Br-5-OCH_3_ (Fig. 1E and 1F). By contrast, Arg 184 was predicted to form a hydrogen-bond interaction with oxygen in the -OCF_3_ moiety of 3-Cl-5-OCF_3_. In 3-Cl-5-CF_3_, the absence of oxygen precluded the formation of the above-mentioned hydrogen bond between Arg 184 and the -CF_3_ group, although Arg 184 was still in proximity to the -CF_3_ group. Finally, in 3-Br-5-OCF_3_, Arg 184 was predicted to form a hydrogen bond with the oxygen atom in the -OCF3 group, as in 3-Cl-5-OCF_3_.

The effects of these NR4A1 ligands and NR4A1-dependent transactivation were determined in MDA-MB-231 breast cancer cells transfected with constructs containing a yeast GAL4-NR4A1 chimera and a UAS-luc reporter gene (containing five GAL4 response elements) (Figure 2A) and an NBRE-luc construct containing a monomeric NBRE site that binds monomeric NR4A (Figure 2B). Treatment with the 3,5-disubstitued analogs decreased transactivation in MDA-MB-231 cells. A similar response was observed in SKBR3 cells transfected with GAL4-NR4A1/UAS-luc (Figure 2C) or NBRE-luc (Figure 2D) constructs and treated with the 3,5-disubstituted analogs. These results show that the 3,5-disubstituted phenyl analogs act as NR4A1 antagonists in breast cancer cells, as previously observed with other 3,5-disubstituted phenyl bis-indoles containing the 4-hydroxyl group [20,21].

### 3.2. In Vitro Effects of 3,5-Disubstituted-Phenyl CDIMs

Treatment of MDA-MB-231 cells with 5 and 10 µM 3-Br-5-OCH_3_ (Figure 3A), 3-Cl-5-OCF_3_ (Figure 3B), 3-Cl-5-CF_3_ (Figure 3C), and 3-Br-5-OCF_3_ (Figure 3D) inhibited cell growth and these growth-inhibitory effects were significantly attenuated in cells where NR4A1 was decreased by RNAi. Results of the NR4A1 knockdown are illustrated in Figure 3E. Western blot analysis of whole cell lysates from MDA-MB-234 cells treated with 5 or 10 µM concentrations of these compounds decreased the expression of several NR4A1-regulated gene products, including EGFR, cMET phosphorylated mTOR, and PD-L1 that are inhibited by a CDIM/NR4A1 antagonist [18,20] (Figure 3F). In addition, we also observed a decreased expression of the mTOR and NR4A1 proteins for some concentrations of these compounds in MDA-MB-231 cells. Treatment with 5–10 µM 3-Br-5-OCH_3_, 3-Cl-5-OCF_3_, 3-Cl-5-CF_3_, and 3-Br-5-OCF_3_ (Figure 4A–D, respectively) inhibited the growth of SKBR3 breast cancer cells and, treatment-related growth and inhibition were significantly attenuated after the NR4A1 knockdown by RNAi (Figure 4E illustrates the effectiveness of RNAi in decreasing NR4A1 protein). These 3,5-disubstituted CDIMs (5 or 10 µM) also decreased the expression of EGFR, cMET, and phosphorylated mTOR (Figure 4F) as observed in MDA-MB-231 cells (Figure 3F). PD-L1 is not expressed in this cell line and the compound-induced downregulation of mTOR and NR4A1 was less pronounced or not observed for all these compounds, with the exception of 3-Br-5-OCF_3_, which was the most potent 3,5-disubstituted CDIM compound with respect to antagonizing the expression of NR4A1-regulated gene products.

### 3.3. In Vivo Antitumorigenic Activities of 3,5-Disubstituted-Phenyl CDIMs

In previous studies, the most active buttressed compound of CDIM8 (i.e., containing the 4′-hydroxyl group) was the 3′-chloro-4-methoxy analog, which significantly inhibited mammary growth (<50%) at a dose of 2 mg/kg/d [21,22] in athymic nude mice bearing MDA-MB-231 cells (orthotopic). In this study, we used the same mouse model and examined the tumor growth-inhibitory effects of the four 3,5-disubstituted phenyl NR4A1 antagonists at a dose of 1 mg/kg/d. Results in Figure 5A–D show that 3-Br-5-OCH_3_ decreased the size of tumors compared to controls, did not affect body weight but decreased tumor growth during the three-week treatment, which resulted in decreased tumor weights, respectively. Similar effects were observed in mice treated with 3-Cl-5-OCF_3_, tumor size was decreased (Figure 5E), body weights were unchanged (Figure 5F), tumor volumes were decreased over the treatment period (Figure 5G), and tumor weights were decreased compared to corn-oil treatment controls (Figure 5F).

3-Cl-5-CF_3_ (1 mg/kg/d) also decreased tumor size (Figure 6A), did not affect body weights (Figure 6B), and decreased tumor volumes (Figure 6C) and tumor weights (Figure 6D). Similar results were observed in mice treated with 3-Br-5-OCF_3_ where tumor size was unchanged (Figure 6E), body weights were not affected (Figure 6F), and tumor volumes (Figure 6G) and weights (Figure 6H) were decreased. These results clearly demonstrate the potency of the 3,5-disubstituted phenyl bis-indole derivatives as tumor growth inhibitors, which are significantly more potent than CDIM8 and other 4-substituted phenyl analogs. We also observed that in tumor lysates from the treated mice that the NR4A1-regulated gene products EGFR, cMET, PD-L1, and phosphorylated mTOR were decreased (Figure 7A,B) and that these in vivo results complement the in vitro studies in MDA-MB-231 cells (Figure 3).

## 4. Discussion

The nuclear receptors NR4A1, NR4A2 (Nurr1), and NR4A3 (Nor1) respond to multiple cellular stressors and play an important role in maintaining cellular homeostasis and pathophysiology [34,35]. The NR4A subfamily play varied roles in cancer and have been characterized as both tumor promoters and tumor suppressors, depending on the tumor type. For example, the double knockout NR4A1^−/−^/NR4A3^−/−^, mice rapidly develop acute myelocytic leukemia (AML) and both receptors exhibit tumor suppressor-like activity as in most blood-derived cancers [36,37]. In contrast, NR4A1 is overexpressed in most solid tumor-derived cancers, including breast cancer, and results of knockdown studies demonstrate the role of NR4A1 in regulating pathways and genes associated with cancer cell proliferation, survival, and migration/invasion [18]. The NR4A subfamily are orphan nuclear receptors with no known endogenous ligands; however, for NR4A1 structurally diverse agents indirectly or directly modulate NR4A1-regulated pathways. Many apoptosis-inducing agents induce cell death in cancer cells through the induction of the nuclear export of NR4A1, which can form a pro-apoptotic complex with bcl-2 to disrupt mitochondria and induce apoptosis [38,39]. Wu and coworkers have shown that cytosporone B and related compounds bind NR4A1 and can act through both nuclear and nuclear export pathways [40,41].

A series of bis-indole-derived analogs (CDIMs) that bind NR4A1 were developed in this laboratory, and initial studies focused primarily on the p-hydroxyphenyl (CDIM8) and p-carboxymethyl phenyl analogs, which acted as NR4A1 antagonists and inactivated NR4A1-regulated pro-oncogenic pathways and genes in cancer cells. These compounds inhibited tumor growth in athymic nude mice bearing breast and other cancer cells at doses of 20–40 mg/kg/d [13,20,25,26,27]. The potency of CDIM8 was limited due to rapid metabolism at the hydroxyl group (conjugation) [23]. In contrast, a series of buttressed analogs of CDIM8 containing 3′- and 5′-substituents, which hinder the metabolism of the hydroxy group exhibited anticancer activity at lower doses than observed for CDIM8. One of these compounds 1,1-bis(3′indolyl)-1-(3-chloro-4-hydroxy-5-methoxyphenyl)methane inhibited breast tumor growth in athymic and immunocompetent mouse models at doses in the 2–3 mg/kg/d range [21,22]. Our previous studies assumed the requirement of the 4′-hydroxyl group for the activity of CDIMs as NR4A1 ligands; however, the receptor-binding results with bis-indole-derived CDIMs containing two or three substituents on the phenyl ring indicated that the 4-hydroxyl group was not required for binding NR4A1 and that the most avid ligands for binding NR4A1 were 3,5-disubstituted phenyl compounds. In this study, we selected four 3,5-disubstituted CDIM compounds which exhibited the lowest K_D_ values for binding NR4A1 (Figure 1) and investigated their activity as antagonists of NR4A1-regulated pro-oncogenic pathways/gene products in MDA-MB-231 and SKBR3 breast cancer cells. These cell lines were selected based on their use in prior studies of 3,5-disubstituted 4-hydroxyphenyl CDIMs [22], and therefore potencies of the new 3,5-disubstituted-phenyl analogs could be directly compared to other ligands.

All four compounds inhibited NR4A1-dependent transactivation and the growth of MDA-MB-231 and SKBR3 cells, which was accompanied by the downregulation of EGFR, cMET, and phosphorylated mTOR. These results are consistent with studies on CDIM8 and buttressed analogs of CDIM8 in breast and other cancer cell lines [19,20,21,22,23,24]. The data indicate that the 3,5-disubstituted compounds were NR4A1 antagonists and we hypothesized that their in vivo activity as inhibitors of tumor growth would be similar to or greater than the buttressed analogs of CDIM8. The results (Figure 5, Figure 6 and Figure 7) indicate that 3-Br-5-OCH_3_, 3-Cl-5-OCF_3_, 3-Cl-5-CF_3_, and 3-Br-5-OCF_3_ inhibited tumor growth at a dose of 1 mg/kg/d. We assume that the high potency of these compounds is due to their high-binding affinity for NR4A1 and also to the absence of the 4-hydroxyl group on the phenyl ring, which decreases conjugation. This is being investigated further in pharmacokinetic studies. The results of the in vivo studies indicate that the IC_50_ for tumor growth inhibition is <1 mg/kg/day for the 3,5-disubstituted CDIM compounds. This potency is greater than that observed for paclitaxel in the same animal model where >50% tumor growth inhibition was observed at a dose of 10 mg/kg (2× weekly) [42]. Interestingly, the in vitro inhibition of MDA-MB-231 cell growth by paclitaxel (200 nM) and 3,5-disubstituted CDIMs (5–10 µM) was inversely related to the higher potency of the CDIMs as compared to paclitaxel as tumor growth inhibitors, and this may be related to lower rates of metabolism.

In summary, our results demonstrate that the 3,5-disubstituted CDIM compounds used in this study are potent inhibitors of breast cancer cell and tumor growth and represent a highly potent class of NR4A1 ligands. It is likely that the anticancer activities of these compounds are due, in part, to their tissue persistence and this will be investigated in future studies. In addition, we are also carrying out isothermal calorimetry binding assays and computer-based ligand-NR4A1 modeling to further probe ligand–receptor interactions and the development of the NR4A1 ligands with enhanced binding affinities. These assays, coupled with additional in vivo studies, will be used to select lead compounds for future preclinical and clinical applications.

## Figures and Tables

**Figure 1 cancers-13-02682-f001:**
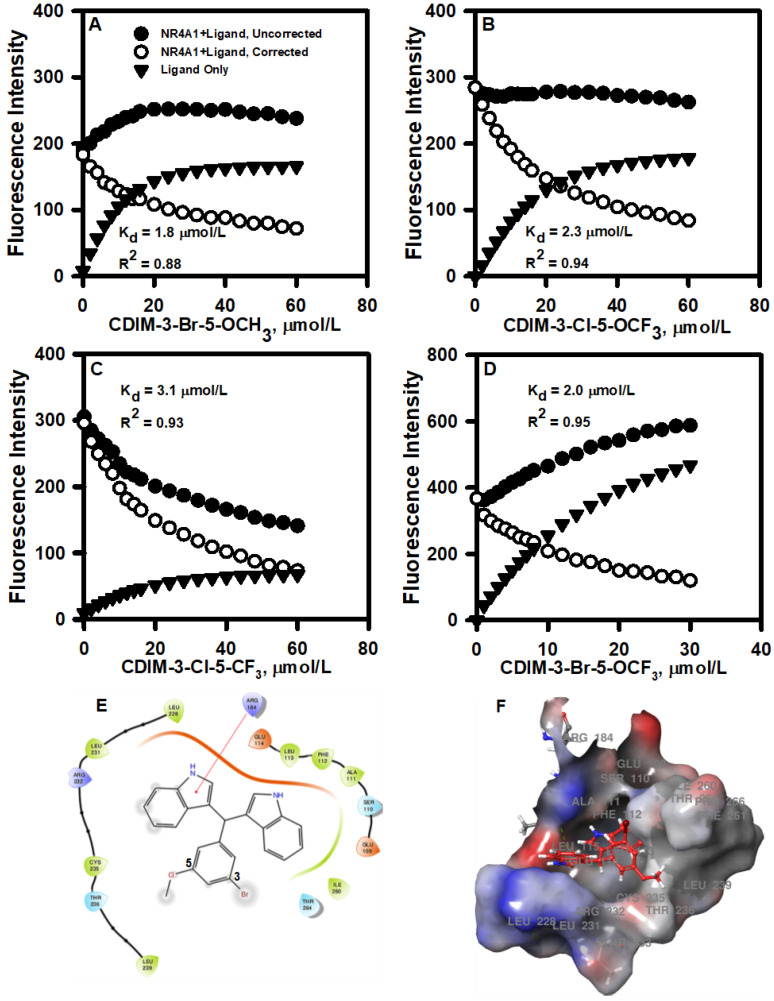
3,5-Disubstituted CDIMs as NR4A1 ligands: Direct binding of the 3-Br-5-OCH_3_ (**A**), 3-Cl-5- OCF_3_ (**B**), 3-Cl-5- CF_3_ (**C**), and 3-Br-5- OCF_3_ (**D**). Phenyl-substituted CDIM compounds with the ligand-binding domain (LBD) of NR4A1 was determined by the fluorescent quenching of a Trp residue in the LBD (19). Example of 3-Br-5-OCH_3_ interactions with amino acid side chains (**E**) and the binding pocket of NR4A1 (**F**).

**Figure 2 cancers-13-02682-f002:**
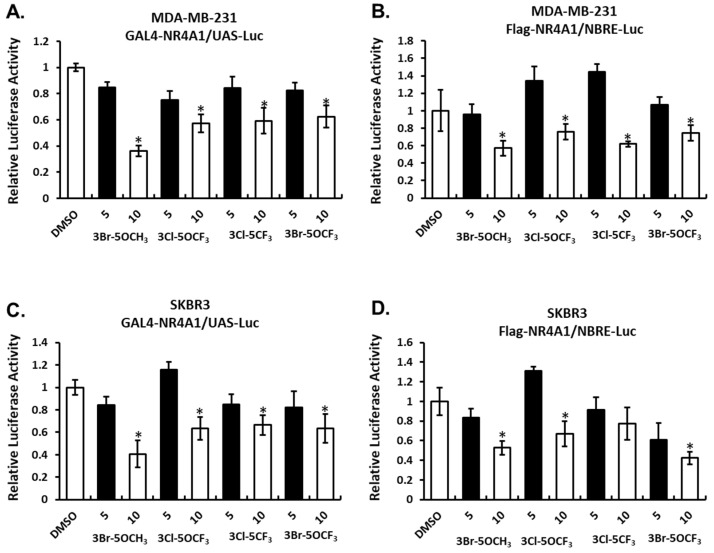
Compound-mediated NR4A1-dependent transactivation in breast cancer cells. MDA-MB-231 cells were transfected with GAL4-NR4A1/UAS-luc (**A**) or Flag NR4A1/NBRE-luc (**B**) constructs, treated with CDIM compounds and luciferase activity determined, as outlined in the Materials and Methods section. CDIM/NR4A1 antagonist-mediated luciferase activity in SKBR3 cells transfected with GAL4-NR4A1/UAS-luc (**C**) and Flag NR4A1/NRBE-luc (**D**) was determined, as described for MDA-MB-231 cells. Results are expressed as means ± SD for at least three replicate determinations for each concentration, and significant (*p* < 0.05) inhibition is indicated (*).

**Figure 3 cancers-13-02682-f003:**
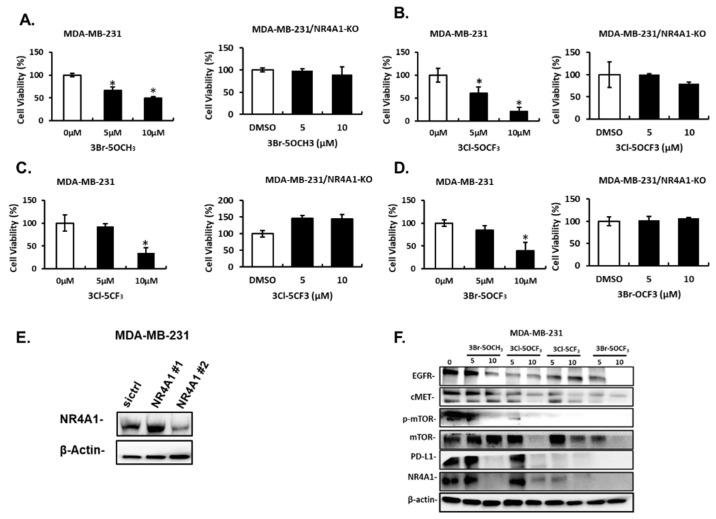
Effects of 3,5-disubstituted phenyl CDIM analogs on MDA-MB-231 cell viability and NR4A1-regulated gene products. MDA-MB-231 cells were treated with 3-Br-5-OCH_3_ (**A**), 3-Cl-5- OCF_3_ (**B**), 3-Cl-5- CF_3_ (**C**), and 3-Br-5- OCF_3_ (**D**) for 24 h and effects on cell proliferation were determined in cells expressing NR4A1 and after NR4A1 knockdown, as outlined in the Materials and Methods section (**E**), The uncropped Western blots have been shown in Appendix A. Cells were transfected with oligonucleotides targeting NR4A1 and after 72 h whole cell lysate were analyzed by western blots (**F**). (NR4A1 #2 used above) Cell lysates from MDA-MB-231 cells were treated with the CDIM compounds and analyzed by western blots to determine effects on NR4A1-regulated gene products. Results (**A**–**D**) are given as means ± SD for at least three replicate determinations for each treatment group, and significant (*p* < 0.05) inhibition is indicated (*).

**Figure 4 cancers-13-02682-f004:**
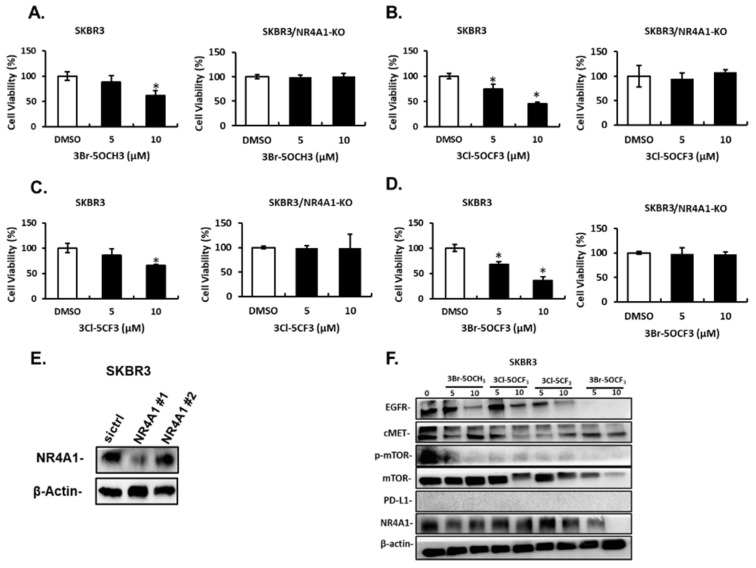
Effects of 3,5-disubstituted phenyl CDIM analogs on SKBR3 cell viability and NR4A1-regulated gene products. SKBR3 cells were treated with 3-Br-5-OCH_3_ (**A**), 3-Cl-5- OCF_3_ (**B**), 3-Cl-5- CF_3_ (**C**), and 3-Br-5- OCF_3_ (**D**) for 24 h and effects on cell proliferation were determined in cells expressing NR4A1 and after the NR4A1 knockdown, as outlined in the Materials and Methods section (**E**). Cells were transfected with oligonucleotides targeting NR4A1, and after 72 h whole cell lysate were analyzed by western blots (**F**). (NR4A1 #1 used above) Cell lysates from SKBR3 cells were treated with the CDIM compounds and analyzed by western blots to determine effects on NR4A1-regulated gene products. Results (**A**–**D**) are given as means ± SD for at least three replicate determinations for each treatment group, and significant (*p* < 0.05) inhibition is indicated (*).

**Figure 5 cancers-13-02682-f005:**
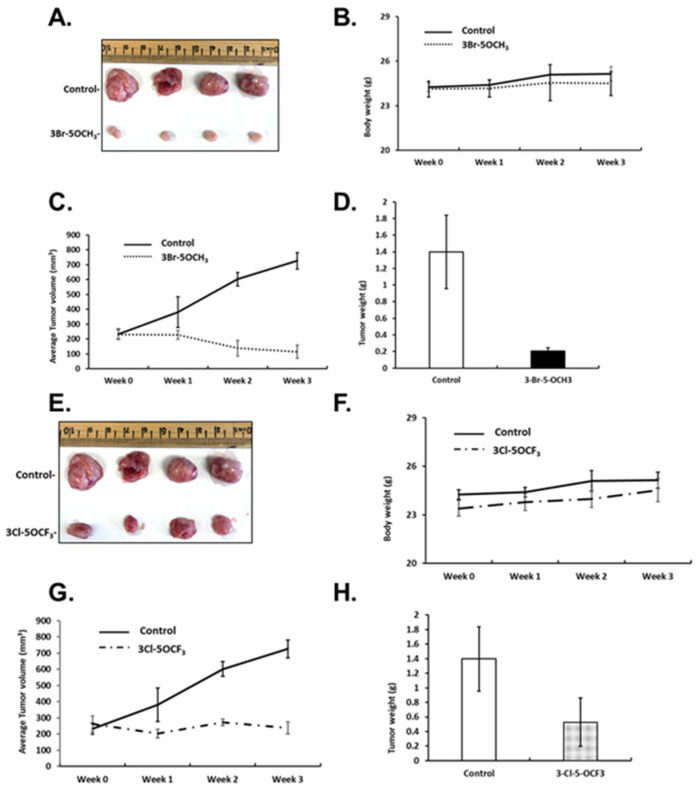
Inhibition of tumor growth by 3-Br-5-OCH_3_ and 3-Cl-5-OCF_3_ phenyl-substituted CDIMs. Athymic nude mice were injected with MDA-MB-231 cells orthotopically and administered corn oil or 3-Br-5-OCH_3_ (1 mg/kg) daily for 21 days and effects on tumor volume (**A**), body weight (**B**), tumor growth (**C**), and weight (**D**) were determined, as outlined in the Materials and Methods section. The same protocol was used for 3-Cl-5-OCF_3_ (1 mg/kg/d) phenyl-substituted CDIM and effects on tumor volume (**E**), body weight (**F**), tumor growth (**G**), and weight (**H**) were determined.

**Figure 6 cancers-13-02682-f006:**
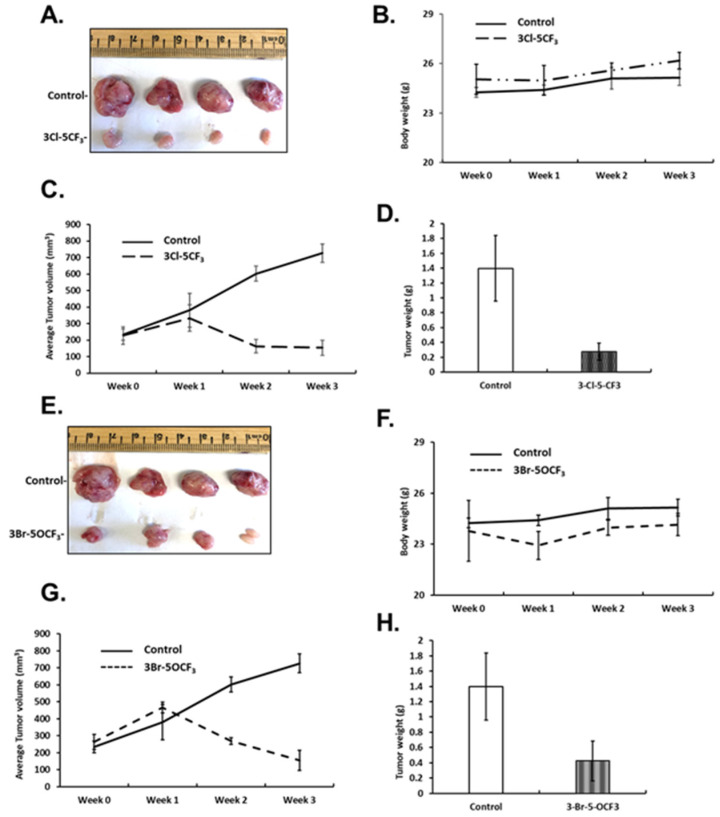
Inhibition of tumor growth by 3-Cl-5-CF_3_ and 3-Br-5-OCF_3_ phenyl-substituted CDIMs. Athymic nude mice were injected with MDA-MB-231 cells orthotopically and administered corn oil or 3-Cl-5-CF_3_ (1 mg/kg) daily for 21 days. Effects on tumor volume (**A**), body weight (**B**), tumor growth (**C**), and weight (**D**) were determined, as outlined in the Materials and Methods section. The same protocol was used for 3-Br-5-OCF_3_ (1 mg/kg/d) phenyl-substituted CDIM and effects on tumor volume (**E**), body weight (**F**), tumor growth (**G**), and weight (**H**) were determined.

**Figure 7 cancers-13-02682-f007:**
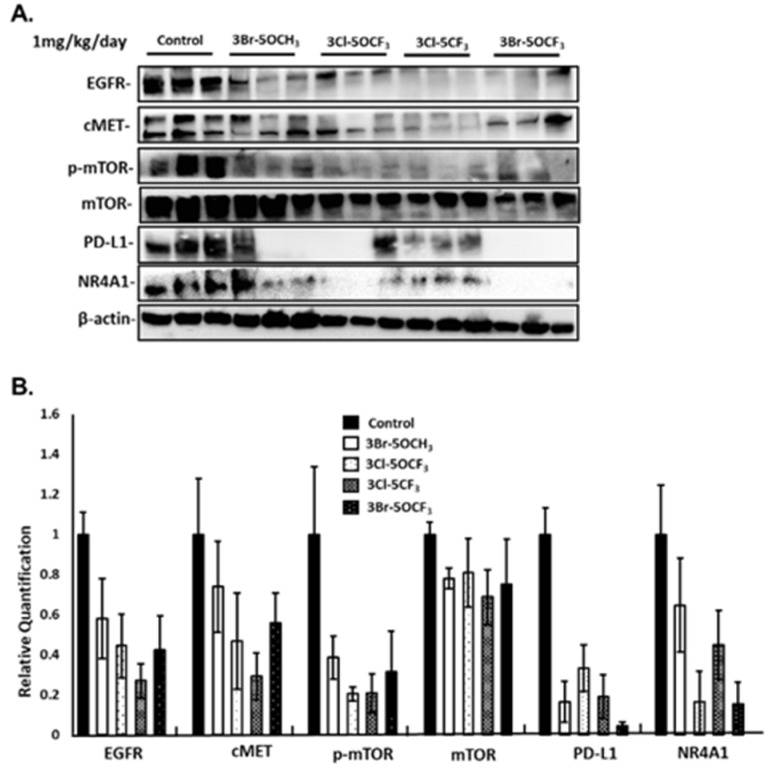
In vivo studies: analysis and quantitation of gene products. (**A**) Tumor lysates from the in vivo studies (Figure 5 and Figure 6) were analyzed by western blots and band intensities were quantitated (**B**) relative to β-actin, as outlined in the Materials and Methods section.

## Data Availability

All data generated presented in this paper and raw data are archived by Stephen Safe and Texas A&M University.

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
