# Peer review of "NR4A1 Ligands as Potent Inhibitors of Breast Cancer Cell and Tumor Growth"

_cancers, 2021, doi:10.3390/cancers13112682_

Round 1

Reviewer 1 Report

An interesting original article about the use of four ligands of Nuclear receptor 4A1 in the treatment of breast cancer. 

The results showed that all four compounds inhibited tumor growth in a mouse model: I have only minor queries:

In 2.10 statistical analysis section, please specify what statistical informatic program was used to calculate significance, its maker, and location.

Page 1 line 37 you should add: "Hormone receptor-positive and human epidermal growth factor receptor 2-negative breast cancer represents
the largest subtype of this neoplastic disease." and cite an article such as: doi: 10.1007/s40264-021-01071-1.

A conclusion paragraph should be added discussing future developemnts of this study's results.

Thank You

Author Response

  1. In 2.10 statistical analysis section, please specify what statistical informatic program was used to calculate significance, its maker, and location.

Response: The Statistical Analysis sections has been revised.

  1. Page 1 line 37 you should add: "Hormone receptor-positive and human epidermal growth factor receptor 2-negative breast cancer represents
    the largest subtype of this neoplastic disease." and cite an article such as: doi: 10.1007/s40264-021-01071-1.

Response: The statement on EGFR and the reference have now been added to the text.

  1. A conclusion paragraph should be added discussing future developments of this study's results.

Response: A concluding paragraph has now been added to summarize results and indicate future directions.

Reviewer 2 Report

The authors show very nicely that breast cancer cell lines expressing Nur77 are highly susceptible to indole analogs that inactivate the Nur77 pathway. They employ synthetic SAR type work with good NMR and computational modeling coupled with Nur77 binding assays and cell cytotoxicity studies. They also include a proof of concept mouse study with orthotopically implanted breast cancer cells and show that the compounds they call leads actually and potent inhibitors of cell growth. A couple of suggestions:

  1. The method to evaluate ligand binding and Kd comes from using fluorescence quenching of tryptophan at the LBD +/_ indole analogs. It is unclear if the compounds themselves have tryptophan quenching ability given that they are indoles and derived off tryptophan. These data should be presented at least in the supplement.
  2. Some off-target related Nurr-type receptor activity should also be investigated --are these compounds specific to Nurr77 and no other target? Alternatively, Nur77 knockdowns (if feasible) should be used for controls and/or Nurr77 compound resistant mutants (if these can be generated) and overexpressed. At the very least some discussion of appropriate controls in the discussion should be stated even if experiments are planned for the future.
  3. The potency in vitro KD < 3 microMolar somehow doesn't match the very significant in vivo effects of the compounds ( 1 mg/kg) which is very dramatic-- concerns here are could the compound be metabolized to other products with other targets? If there is no data at least these should be addressed in the discussion.

Other minor corrections:

  1. Larger and clearer figures would be helpful (e.g., Figure 1 is hard to read).
  2. In the statistics, please state the exact type of significance test used in the legends if possible. The ANOVA is usually for more than two groups tested simultaneously. 
  3. Where possible shoe raw data points like a scatter plot -- but understood if in some figures this might be difficult.
  4. Please state how many samples and repeats were done for each experiment.

Author Response

  1. The method to evaluate ligand binding and Kd comes from using fluorescence quenching of tryptophan at the LBD +/_ indole analogs. It is unclear if the compounds themselves have tryptophan quenching ability given that they are indoles and derived off tryptophan. These data should be presented at least in the supplement.

Response: In previous binding studies the binding of some CDIMs could not be determined due to their quenching ability and a reference to the previous study has now has been included. The compounds use in this study did not interfere with the fluorescence quenching assays and this is now noted and referenced.

  1. Some off-target related Nurr-type receptor activity should also be investigated --are these compounds specific to Nurr77 and no other target? Alternatively, Nur77 knockdowns (if feasible) should be used for controls and/or Nurr77 compound resistant mutants (if these can be generated) and overexpressed. At the very least some discussion of appropriate controls in the discussion should be stated even if experiments are planned for the future.

Response: Results of this study showed that the 3,5-disubstituted CDIMs target many of the same gene products observed after NR4A1 knockdown. In addition, Figures 3 and 4 illustrate new data which show that after NR4A1 knockdown by RNA interference in MDA-MB-231 and SKBR3 cells the growth inhibitory effects of the CDIM analogs were significantly attenuated. This new data demonstrates an important role for NR4A1 in mediating the cytotoxic effects of the CDIM compounds.

  1. The potency in vitro KD < 3 microMolar somehow doesn't match the very significant in vivo effects of the compounds (1 mg/kg) which is very dramatic-- concerns here are could the compound be metabolized to other products with other targets? If there is no data at least these should be addressed in the discussion.

Response: We have now added a summary paragraph that addresses the issues of potency which are probably due, in part, to favorable PK and we are now developing new binding assay (isothermal calorimetry) which may give lower KD values than observed in the fluorescence binding assay. These issues and additional modeling studies will be carried out in future studies in order to develop optimal compound(s) for preIND studies and these issues are addressed in a final conclusion section of the Discussion.

Other minor corrections:

  1. Larger and clearer figures would be helpful (e.g., Figure 1 is hard to read).

Response: We have tried to make the figures more readable.

  1. In the statistics, please state the exact type of significance test used in the legends if possible. The ANOVA is usually for more than two groups tested simultaneously. 

Response: Analysis section has now been revised to address this issue.

  1. Where possible shoe raw data points like a scatter plot -- but understood if in some figures this might be difficult.

Response: For most of the studies the variability in data for individual studies was low and therefore was presented as a bar graphs showing means ± SDs.

  1. Please state how many samples and repeats were done for each experiment.

Response: This is now noted under statistical analysis.

Round 2

Reviewer 2 Report

My comments have been adequately answered.